# In Situ Raman Microdroplet Spectroelectrochemical Investigation of CuSCN Electrodeposited on Different Substrates

**DOI:** 10.3390/nano11051256

**Published:** 2021-05-11

**Authors:** Zuzana Vlčková Živcová, Milan Bouša, Matěj Velický, Otakar Frank, Ladislav Kavan

**Affiliations:** J. Heyrovský Institute of Physical Chemistry, Czech Academy of Sciences, Dolejškova 2155/3, 182 23 Prague, Czech Republic; milan.bousa@jh-inst.cas.cz (M.B.); matej.velicky@jh-inst.cas.cz (M.V.); otakar.frank@jh-inst.cas.cz (O.F.); ladislav.kavan@jh-inst.cas.cz (L.K.)

**Keywords:** CuSCN, hole-transport material, carbon, in situ Raman spectroelectrochemistry

## Abstract

Systematic in situ Raman microdroplet spectroelectrochemical (Raman-μSEC) characterization of copper (I) thiocyanate (CuSCN) prepared using electrodeposition from aqueous solution on various substrates (carbon-based, F-doped SnO_2_) is presented. CuSCN is a promising solid p-type inorganic semiconductor used in perovskite solar cells as a hole-transporting material. SEM characterization reveals that the CuSCN layers are homogenous with a thickness of ca. 550 nm. Raman spectra of dry CuSCN layers show that the SCN^−^ ion is predominantly bonded in the thiocyanate resonant form to copper through its S−end (Cu−S−C≡N). The double-layer capacitance of the CuSCN layers ranges from 0.3 mF/cm^2^ on the boron-doped diamond to 0.8 mF/cm^2^ on a glass-like carbon. In situ Raman-μSEC shows that, independently of the substrate type, all Raman vibrations from CuSCN and the substrate completely vanish in the potential range from 0 to −0.3 V vs. Ag/AgCl, caused by the formation of a passivation layer. At positive potentials (+0.5 V vs. Ag/AgCl), the bands corresponding to the CuSCN vibrations change their intensities compared to those in the as-prepared, dry layers. The changes concern mainly the Cu−SCN form, showing the dependence of the related vibrations on the substrate type and thus on the local environment modifying the delocalization on the Cu−S bond.

## 1. Introduction

The architecture of both solid-state dye-sensitized solar cell (ss-DSSC) [1,2,3] and perovskite solar cell (PSC) [4,5] requires the hole-transport layer, conducting the photo-generated holes to a positive terminal of the cell. The advantages of ss-DSSCs over the traditional liquid-junction DSSCs are (i) prevention of dye desorption, (ii) absence of liquid leakage or evaporation and (iii) a facile device encapsulation and stacking in series [3]. The first materials that made it possible to replace liquid electrolytes were organic hole-transport materials: triphenyl diamine with a hole mobility of about 10^−^^3^ cm^2^V^−^^1^s^−^^1^ [6] and 2,2′,7,7′-tetrakis-(N,N-di-p-methoxyphenylamine)-9,9′-spirobifluorene (spiro-OMeTAD) with a mobility of the amorphous phase of ~10^−^^5^ cm^2^V^−^^1^s^−^^1^ [7]. Another very promising solid p-type inorganic semiconductor, used in photovoltaic devices (ss-DSSC [8,9,10,11,12,13,14,15] and PSC [5,16,17,18,19]) as a hole-transport material, is a copper(I) thiocyanate (CuSCN). Due to its advantageous electrical properties (wide bandgap of ~3.9 eV, hole mobility 0.001–0.1 cm^2^V^−^^1^s^−^^1^) [20,21,22], chemical stability and optical transparency [23], CuSCN is a good alternative to the more expensive and less-stable spiro-OMeTAD particularly if CuSCN is interfaced to a layer of reduced graphene oxide [19,24]. This finding illustrates that the CuSCN/carbon junction is of primary interest for applications in photovoltaics.

The SCN^−^ ion may coordinate to a metal atom (*M*) through either the S or the N donor sites. As a result, thiocyanate (M−S−C≡N), isothiocyanate (M−N=C=S), or bridging (M–SCN–M’) linkage isomers are possible [25,26]. Electrodeposition from an aqueous solution containing SCN^−^ and Cu^2+^ ions is one of the low-temperature deposition techniques for CuSCN layers [23,27,28,29,30,31,32,33,34] in addition to spin-coating [24], drop-casting [35] or spray deposition [36]. This technique allows the CuSCN layer deposition on conductive substrates, such as the commonly used conducting glass (Fluorine doped Tin Oxide/FTO, or Indium Tin Oxide/ITO), carbon substrates (glass-like carbon/GC, boron-doped diamond/BDD, highly ordered pyrolytic graphite/HOPG), or metals. Since these photovoltaic systems are composed of several individual layers (sandwich architecture), which form interface transitions, it is necessary to know the materials’ structural changes at different applied voltages to improve device stability and efficiency.

In situ Raman spectroelectrochemistry (SEC) is a useful characterization method combining Raman spectroscopy and electrochemistry, which allows monitoring of the structural and electronic changes of the studied material with the applied potential. Raman spectra (changes in frequencies and intensities) are influenced by the double-layer charging (electrochemical doping) and by Faradaic processes on the electrode surface [37]. A commonly used in situ Raman SEC setup is a closed electrochemical cell with both the electrolyte solution and the sample contained inside, which only allows the application of potential across the whole electrochemically active area of the immersed sample. Such a “macro setup” has further disadvantages, such as the attenuation of the Raman intensity due to the optical window and the need for a flawless sealing of the electrical contact to the working electrode. An in situ Raman SEC arrangement using a microdroplet electrochemical cell (Raman-μSEC) can be employed to avoid these drawbacks [38,39,40]. The μSEC allows highly localized electrochemical studies on spatially heterogeneous surfaces (typical microdroplet diameter of 10–20 μm), such as two-dimensional materials [38,39,40], where only a well-defined microscale area of the sample in contact with the microdroplet is affected by the applied potential.

The electrochemical behavior of quasi-metallic or semiconducting carbon materials, such as polycrystalline doped diamond with strong and nearly isotropic tetrahedral sp^3^ bonding, or carbons with sp^2^ hybridization, such as disordered glass-like carbon or highly oriented pyrolytic graphite, are described in the literature [41,42,43,44,45,46,47,48,49]. Compared to sp^2^ materials, doped diamonds have a wider electrochemical potential window in aqueous media and they are (electro)chemically more stable. The first in situ Raman SEC study [45] of polycrystalline boron-doped diamond films with varying sp^2^ carbon content and boron doping performed in a closed electrochemical cell showed the stability of the sp^3^ diamond lattice and incorporated boron atoms, while specific spectral changes of the sp^2^ carbon under cathodic and anodic treatment were observed. The spectroelectrochemical behavior of sp^2^ carbon materials (graphite, graphene, carbon nanotubes, or fullerenes) is accompanied by changes in the Raman frequencies and intensities, described in detail in refs. [37,41,50,51,52].

In our previous work [19], we performed an in situ Raman spectroelectrochemistry of spin-coated CuSCN layers prepared on gold and GC substrates. Herein, we present an in-depth advanced microdroplet SEC investigation of CuSCN layers electrodeposited from an aqueous solution at room temperature on different substrates: (i) carbon-based materials (BDD, GC and HOPG) with varied amounts of surface oxygen groups on sp^2^ carbon and (ii) carbon-free FTO with a high concentration of surface oxygen-containing groups. Such a combination of various substrate properties enables us to study the structural and electrochemical properties of the electrodeposited CuSCN layers in detail.

## 2. Materials and Methods

Two different types of conductive substrates, i.e., carbon-based (BDD, GC, HOPG) and FTO conducting glass, were used to prepare the CuSCN layer by electrodeposition from an aqueous electrolyte solution following the methodology in ref. [23]. Polycrystalline BDD film was deposited on fused silica in an ASTeX 5010 (Seki Technotron, Tokyo, Japan) series microwave plasma-enhanced chemical vapor deposition reactor [45]. The BDD film was grown in a conventional CH_4_/H_2_ plasma and doping was induced by trimethyl boron gas B(CH_3_)_3_. The growth conditions were as follows: B/C ratio in the gas phase 1000 or 2000 ppm (designated as BDD or BDD 2000, respectively), pressure 47.7 mbar, temperature 720 °C, methane content 0.8% and deposition time 60 min. Glass-like carbon (GC 3000C) with a resistivity of 400 μOhm cm was purchased from Good fellow Cambridge Ltd. (Huntingdon, UK). HOPG substrate was prepared as a thin sheet, peeled off using an adhesive Scotch^®^ double sided tape (3M Company, Maplewood, MN, USA) from the bulk crystal (12 × 12 × 2 mm^3^, ZYB Grade, Momentive Performance Materials Quartz, Inc., Strongsville, OH, USA) and glued onto the glass. FTO with the resistivity of 15 Ohm/sq (TEC 15, Libbey-Owens-Ford Glass Co., Toledo, OH, USA) was used as a carbon-free substrate. The electrodeposition of CuSCN layers was performed potentiostatically at a potential of −0.5 V vs. Ag/AgCl (3 M KCl) for 30 min in a three-electrode cell at room temperature. In this arrangement, the substrate was the working electrode and a Pt mesh was used as the counter electrode. The electrolyte solution was mixed from cupric sulfate pentahydrate (99.9%, Sigma-Aldrich, St. Louis, MO, USA) and potassium thiocyanate (99.0%, Sigma-Aldrich) as precursors and triethanolamine (TEA; 99.0%, Sigma-Aldrich) as a chelating reagent for Cu(II) cations. The molar ratio [Cu^2+^]:[TEA] was 1:10 and the concentration of Cu^2+^ was 0.01 M. Before electrodeposition, the CuSO_4_/TEA complex solution was mixed with 0.1 M KSCN solution, stirred for 1 h and then stored for 24 h.

All Raman spectra (ex situ and in situ) were excited using a 514 nm line of an Ar^+^ laser (laser power of 1 mW at the sample) and recorded by a LabRAM HR spectrometer (Horiba Jobin-Yvon, Kyoto, Japan) interfaced with an Olympus microscope (100× objective). The spectrometer was calibrated using the F_1g_ mode of Si at 520.2 cm^−1^. The Raman peaks were fitted using Voigt lineshapes. The surface morphology of the samples was investigated employing field emission scanning electron microscopy (FESEM, S-4800 Hitachi, Tokyo, Japan). SEM images were acquired at an acceleration voltage of 5 kV and a working distance of 9–11 mm. The thickness of the electrodeposited layer on carbon substrates was measured by profilometry (Dektak 150, Veeco Instruments Inc., Plainview, NY, USA) over a scratch made by a glass tip. Cyclic voltammetry (CV) was performed in a closed three-electrode cell under Ar atmosphere (working electrode: CuSCN layer, counter electrode: Pt mesh, reference electrode: Ag/AgCl in 3 M KCl). The electrolyte solution was aqueous 0.5 M KCl (Sigma-Aldrich) saturated with CuSCN (pH 6). Electrochemical measurements were carried out using an Autolab PGSTAT128N potentiostat (Metrohm AG, Herisau, Switzerland) controlled by GPES4 software. In situ SEC characterization was performed in a microdroplet electrochemical cell. The microdroplet of 6 M LiCl aqueous electrolyte solution was expelled through a microcapillary containing the reference (Ag/AgCl wire) and counter (Pt wire) electrodes. The interface between the sample surface and microdroplet serves as the working electrode. Note that high electrolyte concentration prevents water evaporation and reduces solution resistance.

## 3. Results

### 3.1. Structural and Electrochemical Characterization of CuSCN Layers

SEM was used to study the morphological properties of the electrodeposited CuSCN thin layer on various substrates. Figure 1A–D display the plan view SEM micrographs of the dense fine-crystalline morphology of the uniform structure of fabricated CuSCN layers on four different substrates. The thickness of the CuSCN layer deposited on the FTO substrate (CuSCN/FTO) determined from a cross-sectional SEM image is approximately 550 nm, comparable to the thickness of CuSCN layers deposited on carbon-based substrates determined from stylus profilometry (approx. 450 ± 40 nm), when the uncertainty stemming from the surface scratching is taken into account. Surface morphology comparison of the CuSCN layers prepared by spin-coating from diethyl sulfide solution in our previous work [19] and layers prepared by electrodeposition in this work shows that surface morphology of the electrodeposited CuSCN layers differs significantly from layers prepared by spin-coating; the crystal size of the electrodeposited layers is significantly larger with a well-defined orientation. The root-mean-square (RMS) roughness values of individual CuSCN layers were determined using profilometry. As expected, RMS values are higher for CuSCN layers prepared by electrodeposition (RMS increases from 19 nm for CuSCN/BDD to 30 nm for CuSCN/FTO; see Figure 1) compared to the CuSCN layers prepared by spin-coating (RMS for CuSCN/GC is approximately 9 nm [19]).

Ex situ Raman spectra in the range of 100–2400 cm^−1^ of the pristine dry CuSCN layers electrodeposited on various substrates (BDD, GC, HOPG, FTO), as well as spectra of bare substrates with their characteristic Raman features [45,49,53,54] are shown in Figure 2A. Raman spectrum of single-crystal graphite (sp^2^ carbon) exhibits only one narrow peak at ~1580 cm^−1^ (E_2g_ phonon, designated as the G mode) in this range [55], while polycrystalline graphite or amorphous sp^2^ carbon (like GC) have an additional peak at ~1350 cm^−1^ (A_1g_ phonon, D mode) [54], whose intensity reflects the effective crystallite size L_a_ [56]. The Raman spectrum of diamond (sp^3^ carbon) exhibits a narrow natural single line at 1332 cm^−1^ [57] identified as the three-fold degenerate, zone-center optical mode at the Γ point and additional broad, weak-intensity peaks near 1580 cm^−1^, which indicate the presence of a small amount of sp^2^ impurities in the films [58]. Bands corresponding to the vibrations associated with the increased boron concentration in BDD start to appear at ~500 and ~1225 cm^−1^. This is accompanied by an increase in electrical conductivity and transition from semimetallic to metallic character of BDD (B content in diamond lattice > 3 × 10^20^ cm^−3^ [59]). At the same time, the characteristic diamond line at 1332 cm^−1^ downshifts, attenuates and exhibits an asymmetric Fano-like line shape [45,60]. The bands of graphitic or amorphous carbon impurities in BDD (depending on the film quality) are located at 1335 cm^−1^ (D band), 1580 cm^−1^ (G band), 1610 cm^−1^ (D band) and 1520 cm^−1^ (tetrahedral amorphous carbon) [45,49]. The Raman bands of pristine FTO are located at 480, 500 and 1080 cm^−1^ [53]. Independently of the substrate type, the Raman spectra of CuSCN layers prepared by electrodeposition exhibit similar Raman features as the CuSCN layers prepared by spin-coating in our previous work [19]. In the higher frequency region (2000–2300 cm^−1^), the electrodeposited CuSCN layers exhibit the peaks of ν_as_(C≡N) stretching mode, which is very sensitive to the local environment of the system as well as to the nature of the cation group [61] in thiocyanate resonant form of the SCN^−^ ion. The weak peak at 2115 cm^−1^ is assigned to the Cu−NCS isomer (Cu^+^−:N≡C−S^−^) and the most intense peak at 2175 cm^−1^ to the Cu−SCN isomer (Cu^+^−^−^S−C≡N). In the spin-coated CuSCN layers from our previous work [19], the most intense peak of the Cu−SCN isomer was red-shifted due to the different morphology of the spin-coated layers [62]. In the low-frequency region of 100–1000 cm^−1^, the presence of the thiocyanate SCN^−^ resonant form is documented by the Raman peaks corresponding to the stretching modes of ν(Cu−S) at 205 cm^−1^, ν(Cu−N) at 241 cm^−1^ and ν(C−S) at 746 cm^−1^ and to the δ(SCN) bending mode at 431 cm^−1^ (indicative of predominant thiocyanate S−binding). The overlapping peaks in the higher-frequency region (same as 746 cm^−1^ peak in the low-frequency region, Appendix A) were fitted with Voigt lineshapes to resolve their complex spectral structure (Appendix A). Additional Raman bands with small intensity appear at ~2159 cm^−1^ corresponding to the bridged –SCN– [26] and at ~2167 cm^−1^ assigned to asymmetric ν(CN) vibration, which shifts due to the varying local environment of the SCN^−^ ion (see below).

Cyclic voltammograms of the CuSCN layers electrodeposited on different substrates (BDD, GC, HOPG, FTO) in 0.5 M KCl sat. CuSCN were recorded over the potential range from −0.3 V to +0.5 V vs. Ag/AgCl to illustrate their electrochemical behavior (Figure 2B). The CVs show the typical behavior of a p-type semiconductor [63]. In the cathodic direction no presence of capacitive charging can be seen, while with the increasing positive potential the anodic current increases due to electrochemical reactions occurring at the electrode surface with variations depending on the substrate type [45]. The double-layer capacitance of CuSCN layers on various substrates determined at 0 V, normalized to the projected geometric surface area, increases from 0.3 mF/cm^2^ for CuSCN/BDD and 0.4 mF/cm^2^ for CuSCN/HOPG to 0.6 mF/cm^2^ for CuSCN/FTO and 0.8 mF/cm^2^ for CuSCN/GC. The capacitances of bare substrates (FTO 30 μF/cm^2^ [19], GC 20 μF/cm^2^ [19], BDD < 12 μF/cm^2^ [45] and HOPG 6 μF/cm^2^ [64]) are at least one order of magnitude lower than the capacitance of the CuSCN layers and therefore do not significantly affect the measurement.

### 3.2. In Situ Raman Microdroplet Spectroelectrochemistry (Raman-μSEC)

A series of in situ Raman spectra of the CuSCN layers electrodeposited on various substrates (FTO, GC, BDD, HOPG) as a function of the applied potential (anodic-black traces, cathodic-gray traces, 0 V vs. Ag/AgCl-pink traces) is shown in Figure 3, Figure 4, Figure 5 and Figure 6 and compared with the (ex situ) Raman spectra of dry CuSCN layers on the respective substrates (bottom spectrum in each figure). The general spectroelectrochemical trends in Raman spectra are comparable with those for the spin-coated CuSCN layers from our previous work, which were measured in the traditional macro-SEC arrangement [19]. A complete attenuation of all Raman vibrations from CuSCN, as well as that of the substrate, was observed in the potential range from 0 V to −0.3 V vs. Ag/AgCl for all four samples. In the potential range from 0 V to +0.5 V vs. Ag/AgCl, the bands corresponding to the CuSCN vibrations change their relative intensities compared to those in the dry layers. The disappearance of the Raman bands at negative potentials is most likely due to the formation of a passivation layer, which changes the surface reflectivity and causes the Raman scattering from bulk phonons to decrease, as observed previously (e.g., for n-GaAs) [65]. In the high-frequency region of 2000–2300 cm^−1^, we observe several types of major changes when approaching +0.5 V. In the case of a CuSCN layer on FTO substrate, three clearly resolved peaks, located at 2159, 2167 and 2175 cm^−1^ are seen in Figure 3E or Figure 7A. While the lowest frequency peak (at 2159 cm^−1^, assigned to the ν(CN) vibration of the -SCN^−^ bridging unit) does not undergo any recognizable change upon the increasing positive potential, the other two high-frequency components switch their relative intensities: the intensity of the peak at 2175 cm^−1^ (ν(CN) vibration of the Cu−SCN isomer) decreases and the intensity of the peak at 2167 cm^−1^ increases. An increase in the 2167 cm^−1^ peak intensity is observed in the case of the BDD substrates also (Figure 4E or Figure 7B for BDD, Appendix A for BDD 2000). An upshift by 2 cm^−1^ is observed in the case of the ν(CN) vibration of the Cu-SCN isomer on GC substrate, however, the changes are not so pronounced. The abovementioned behavior for CuSCN/FTO is in contrast with that of the CuSCN layer grown on the HOPG substrate, for which only a significant narrowing (Δ*FWHM* is 3.8 cm^−1^) of the peak at 2175 cm^−1^ is observed when the potential is increased to +0.5 V. A certain level of linewidth narrowing is present also for the other substrates (Appendix A) and it correlates with their roughness, reflecting the increased sizes of the crystallites.

Two processes affecting the electrodeposited CuSCN layer can thus, be recognized in relation to the potential increase, manifested by the appearance of a new peak at 2167 cm^−1^ (and/or change in relative intensity with the peak of the ν(CN) vibration of the Cu−SCN isomer at 2175 cm^−1^) and the peak narrowing. The latter evidences an increased structural ordering (higher crystallinity and a smaller amount of defects) and/or an increased homogeneity of the crystallites, possibly due to the electrochemical dissolution of smaller CuSCN particles. As for the former, the assignment of the new peak at 2167 cm^−1^ is not straightforward. A clue can be found in the behavior of the lower frequency peaks, especially of the C-S stretching vibration at ~746 cm^−1^, which is also indicative of the Cu−SCN isomer [26,62]. Interestingly, this peak undergoes similar changes as the peak at 2175 cm^−1^, i.e., shifts and appearance of asymmetry are observed. Due to its low intensity, it is impractical to reliably fit the 2167 cm^−1^ peak with more components for a precise comparison with the behavior of the 2175 cm^−1^ peak. Therefore, Appendix A plots the correlation of the Raman shifts of the 746 and 2175 cm^−1^ peak structures, fitted as a single lineshape each. The fitted peak position is then an approximation of the center of mass of the structure. There is indeed a pronounced linear correlation, with the Pearson’s coefficient of 0.8. Hence, we surmise that the ‘new’ peak at 2167 cm^−1^ also belongs to a ν(CN) vibration of the Cu−SCN isomer, where changes in the bond strength take place upon positive bias. The different behavior of the peaks at the particular substrates points to a reorganization of the Cu−SCN units, depending on how strongly the Cu−SCN interacts with the surface—no changes were observed at HOPG, while all the other substrates induced peak shifts and/or intensity changes. The most pronounced changes observed on FTO indicate the oxygen on the surface, which is involved in the charge transfer, can locally modify the delocalization in the Cu−SCN, thereby influencing the molecular vibration. The high sensitivity of the C−N vibration to the charge distribution in the SCN^−^ ion and its surroundings was also observed before [66]. In the case of the carbon substrates, these changes are also influenced by the presence of surface functional groups on the sp^2^ carbon (surface concentration of these groups is very small in HOPG compared to BDD and GC).

The lower frequency regions in Figure 3, Figure 4, Figure 5 and Figure 6 (charts A–C) also exhibit pronounced changes in the positions or intensities of the Cu−S (205 cm^−1^) and Cu−N (241 cm^−1^) peaks. However, these cannot be reliably quantified due to the low intensity and broad nature of the bands. As in the case of spin-coated CuSCN on GC in our previous work [19], a new peak at 470 cm^−1^ appears at positive potentials with the simultaneous disappearance of the ν(SCN) band at 431 cm^−1^ (thiocyanate in the form of an S−bonding) in the case of electrodeposited CuSCN on GC (Figure 5B). We assign the new peak at 470 cm^−1^ to the ν(NCS) bending mode, related to the presence of thiocyanate in the form of an N−bonding [61,67]. Charts D in Figure 3, Figure 4, Figure 5 and Figure 6 illustrate the spectroelectrochemical behavior of CuSCN on each substrate. In our earlier study [45] of bare BDD, we observed relative intensity changes of the sp^2^ carbon modes under the cathodic (intensity increase) and anodic (intensity decrease) treatment. We interpreted these changes as electrochemical charging and/or removal of the sp^2^ carbon surface layers [45]. Interestingly, we do not observe any such changes in the spectroelectrochemical behavior of the BDD substrates below the CuSCN layer (Figure 3D or Appendix A), indicating that the BDD is screened from the electrochemical action. 

## 4. Conclusions

We have studied an inorganic hole-transport material, copper (I) thiocyanate (CuSCN), electrodeposited from aqueous electrolyte solution into a thin layer on two types of conductive/semiconductive substrates; (i) carbon-based substrates (BDD, GC, HOPG) and (ii) carbon-free FTO conducting glass. SEM and Raman characterization evidence that electrodeposition from aqueous solution results in homogenous CuSCN layers (thickness of ca. 550 nm), predominantly in the form of the thiocyanate ion bonded to copper through its S−end (Cu−SCN bonding). The double-layer capacitance of CuSCN ranges from 0.3 mF/cm^2^ (CuSCN/BDD) to 0.8 mF/cm^2^ (CuSCN/GC). We carried out a comparative in situ Raman-μSEC study of CuSCN layers on all four substrates. A complete attenuation of all Raman vibrations from CuSCN, as well as those from the substrate, was observed at negative potentials (0 to −0.3 V vs. Ag/AgCl), caused by the formation of a passivation layer which changes the reflectivity of the surface. At positive potentials (0 to +0.5 V vs. Ag/AgCl), the bands corresponding to the CuSCN vibrations change their intensity compared to those in the dry layers (ex situ). The intensity of the strongest peak of the C≡N stretching vibration at 2175 cm^−1^ decreases and, at the same time, the intensity of another, initially unresolved peak at 2167 cm^−1^ increases. In cases where the individual components at 2175 and 2167 cm^−1^ are not resolved, the change in their relative intensities causes an apparent shift of the broad convoluted peak. The spectral changes are substrate-dependent: the largest changes are observed on FTO, whereas HOPG does not induce any. The shift and intensity variation of the peak composed of the 2175 and 2167 cm^−1^ bands strongly correlate with the shift of the C−S stretching vibration at 746 cm^−1^, thereby suggesting that the mode at 2167 cm^−1^ also originates from the Cu−SCN isomer. The position of this mode is influenced by the substrate, probably through local charges from the oxygen-bearing groups, which cause changes in the delocalization on the C−S bond. We also observed narrowing of the bands upon the application of positive potentials, which reflects an increased ordering of the crystal lattice.

## Figures and Tables

**Figure 1 nanomaterials-11-01256-f001:**
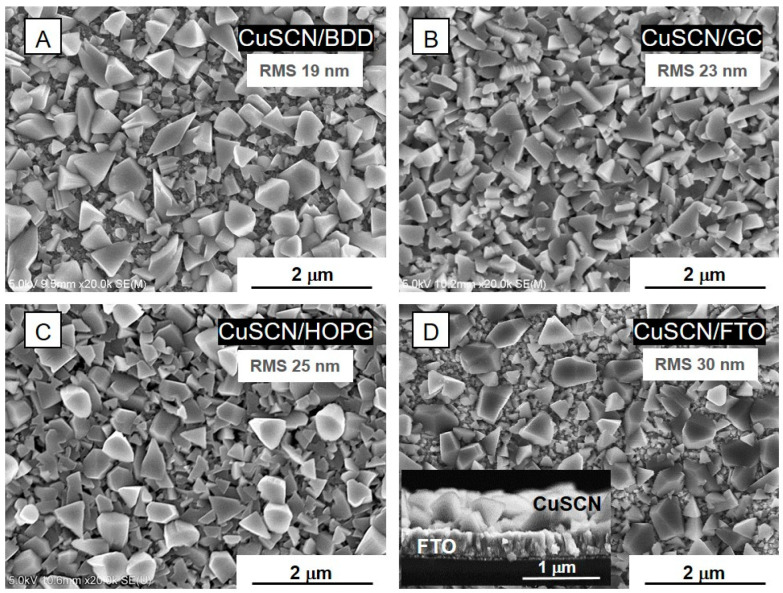
SEM images of CuSCN electrodeposited on (**A**) boron-doped diamond (CuSCN/BDD), (**B**) glass-like carbon (CuSCN/GC), (**C**) highly ordered pyrolytic graphite (CuSCN/HOPG) and D) F-doped SnO_2_ glass (CuSCN/FTO). The inset of chart (**D**) shows the cross-sectional view.

**Figure 2 nanomaterials-11-01256-f002:**
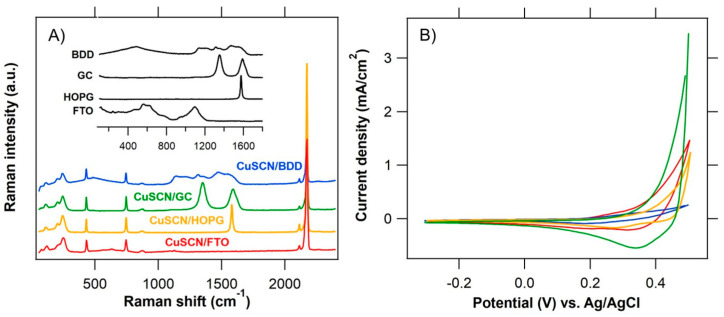
Raman spectra (**A**) and cyclic voltammograms (**B**) of CuSCN layers electrodeposited on (i) boron-doped diamond (CuSCN/BDD, blue trace), (ii) glass-like carbon (CuSCN/GC, green trace), (iii) highly ordered pyrolytic graphite (CuSCN/HOPG, yellow trace) and (iv) FTO glass (CuSCN/FTO, red trace) substrate. The reference Raman spectra of bare substrates are shown in black. The spectra are excited by 514 nm laser radiation and offset for clarity. The electrolyte solution used in chart B was 0.5 M KCl sat. CuSCN, scan rate was 100 mV/s.

**Figure 3 nanomaterials-11-01256-f003:**
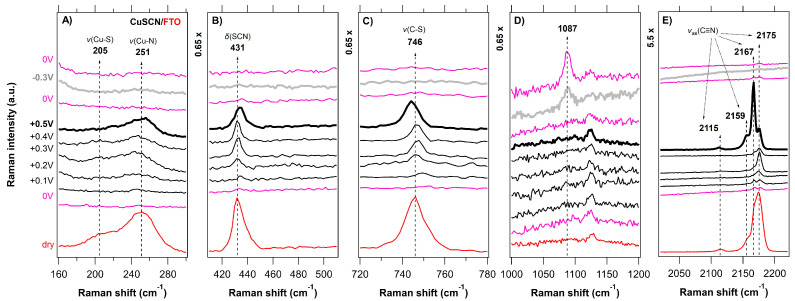
In situ Raman-μSEC spectra of the CuSCN layer (spectral regions of 160–300, 410–510, 720–780, and 2020–2220 cm^–1^ in charts (**A**–**C**,**E**), respectively) electrodeposited on FTO substrate (spectral region of 1000–1200 cm^–1^ in chart (**D**) as a function of the applied potential (vs. Ag/AgCl; shown on the left-side chart) in 6 M LiCl. The measurement sequence was as follows; the first spectrum was acquired at 0 V (bottom pink trace) and the last spectrum was acquired again at 0 V (top pink trace). The sequence of positive potentials (from +0.1 V to +0.5 V) is visualized by black traces and the negative potential of −0.3 V is visualized by the gray trace. The reference (ex situ) Raman spectrum of the dry CuSCN layer on the FTO substrate is shown for comparison (bottom red trace). Spectra are offset for clarity; the intensity scale is identical for all the charts.

**Figure 4 nanomaterials-11-01256-f004:**
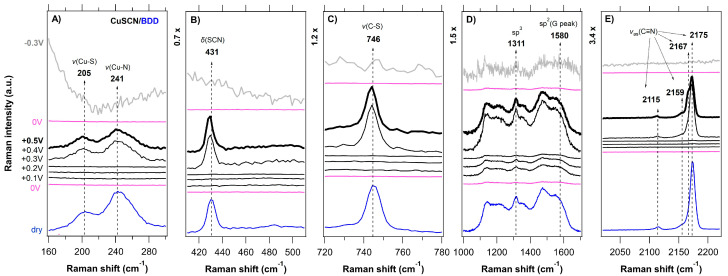
In situ Raman-μSEC spectra of the CuSCN layer (spectral regions of 160–300, 410–510, 720–780, and 2020–2220 cm^–1^ in charts (**A**–**C**,**E**), respectively) electrodeposited on the BDD substrate (spectral region of 1000–1700 cm^–1^ in chart (**D**) as a function of the applied potential (vs. Ag/AgCl; shown on the left-side chart) in 6 M LiCl. The measurement sequence was as follows; the first spectrum was acquired at 0 V (bottom pink trace) and the last spectrum was acquired at −0.3 V (top gray trace). The sequence of positive potentials (from +0.1 V to +0.5 V) is visualized by black traces. The reference (ex situ) Raman spectrum of the dry CuSCN layer on the BDD substrate is shown for comparison (bottom blue trace). Spectra are offset for clarity; the intensity scale is identical for all the charts.

**Figure 5 nanomaterials-11-01256-f005:**
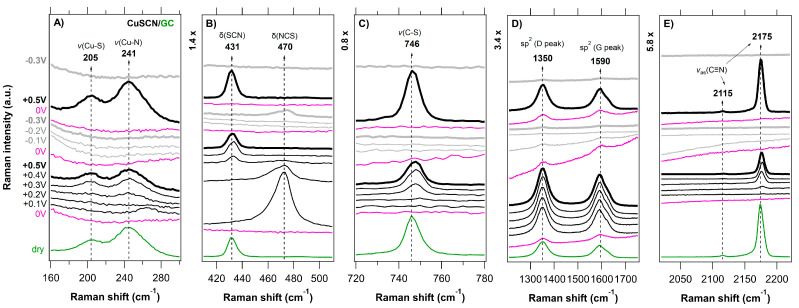
In situ Raman-μSEC spectra of the CuSCN layer (spectral regions of 160–300, 410–510, 720–780, and 2020–2220 cm^–1^ in charts (**A**–**C**,**E**), respectively) electrodeposited on the GC substrate (spectral region of 1210–1750 cm^–1^ in chart (**D**) as a function of the applied potential (vs. Ag/AgCl; shown on the left-side chart) in 6 M LiCl. The measurement sequence was as follows; the first spectrum was acquired at 0 V (bottom pink trace) and the last spectrum was acquired at −0.3 V (top gray trace). The sequence of positive potentials (from +0.1 V to +0.5 V) is visualized by black trace and the negative potentials (from −0.3 V to −0.1 V) are visualized by the gray trace. The reference (ex situ) Raman spectrum of dry CuSCN layer on the GC substrate is shown for comparison (bottom green line). The Electrolyte solution 6 M LiCl. Spectra are offset for clarity; the intensity scale is identical for all the charts.

**Figure 6 nanomaterials-11-01256-f006:**
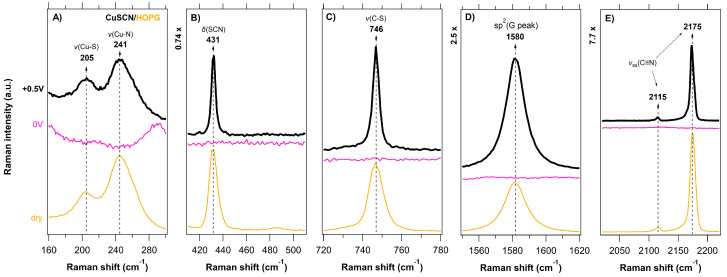
In situ Raman-μSEC spectra of the CuSCN layer (spectral regions of 160–300, 410–510, 720–780, and 2020–2220 cm^–1^ in charts (**A**–**C**,**E**), respectively) electrodeposited on the HOPG substrate (spectral region of 1550–1620 cm^–1^ in chart (**D**) as a function of the applied potential (vs. Ag/AgCl; shown on the left-side chart) in 6 M LiCl. The measurement sequence was as follows; the first spectrum was acquired at 0 V (pink trace) and the last spectrum was acquired at +0.5 V (top black trace). The reference (ex situ) Raman spectrum of dry CuSCN layer on the HOPG substrate is shown for comparison (bottom yellow trace). Spectra are offset for clarity; the intensity scale is identical for all the charts.

**Figure 7 nanomaterials-11-01256-f007:**
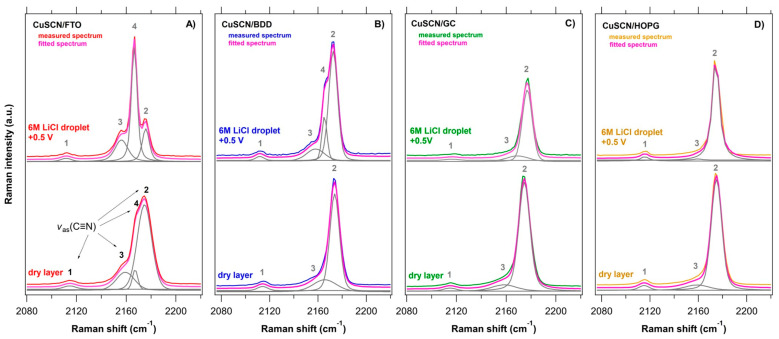
Raman spectra of the electrodeposited CuSCN (i) dry (ex situ) layers (bottom) and (ii) in 6 M LiCl at +0.5 V vs. Ag/AgCl upon in situ Raman-μSEC measurement (top) on (**A**) FTO (red trace), (**B**) GC (green trace), (**C**) BDD (blue trace), (**D**) HOPG (yellow trace) substrates, in the region of SCN vibrations (2080–2200 cm^−1^). The fit envelopes and individual peak components, obtained using Voigt approximation, are shown in pink and gray, respectively.

## Data Availability

The data presented in this study are available on request from the corresponding author.

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
