# Peer review of "In Situ Raman Microdroplet Spectroelectrochemical Investigation of CuSCN Electrodeposited on Different Substrates"

_nanomaterials, 2021, doi:10.3390/nano11051256_

Round 1

Reviewer 1 Report

The authors report a thorough investigation of the CuSCN layers electrodeposited on four different substrates by in situ Raman spectroelectrochemistry. This method revealed a series of findings on the interactions between the selected substrates and the electrodeposited CuSCN layers, including the variable level of ordering of the crystal lattice on the substrate or the influence of oxygen-bearing groups from the substrate on the delocalization of bonding within thiocyanate groups. I have some doubts about the signficance, and thus interest to the readers for such a very specific type of investigation (the authors can comment on the importance of their results in the conclusions to dispel my doubts).

However, the work looks complete and, from this viewpoint I have only two minor points to be considered by the authors:

  1. The authors stated that the thickness of the CuSCN layer deposited on the FTO substrate is ca. 550 nm. The authors should comment about the roughness of this layer as well as about the thickness of other prepared CuSCN layers.
  2. The authors discussed the peak narrowing upon the potential increase and correlates this observation with the increased structural ordering. The authors should comment how this effect is changing upon changing the substrate.

Reviewer 2 Report

This paper presents, to my knowledge a quite original experimental in situ study by Raman scattering of CuSCN electrodes under electric bias voltage.  Raman peaks evolution seems related to structural changes.

To my opinion (I'm not specialist neither of this material nor of Raman scattering), the authors should better explain their observations and the structural consequences they expect (line 289-291  for instance). They should push beyond their interpretation for instance on crystal perfection enhencement. What can lead such things ? Internal stress relief due to charge displacement ? 
